# Post-Pandemic Feeding Patterns and Mediterranean Diet Adherence in Spanish Toddlers

**DOI:** 10.3390/nu15092049

**Published:** 2023-04-24

**Authors:** Ana Isabel Reyes-Domínguez, Javier Bernabeu-Sendra, Cristina Rodríguez-Sinovas, Alicia Santamaria-Orleans, Roser de Castellar-Sanso, Jorge Martinez-Perez

**Affiliations:** 1Pediatric Gastroenterology, Hepatology and Nutrition Unit, Complejo Hospitalario Universitario Insular-Materno Infantil de Gran Canaria, 35016 Las Palmas de Gran Canaria, Spain; 2Pediatric Department, Hospital Clínico Universitario de Valencia, 46010 Valencia, Spain; 3Institut d’Investigació Biomèdica Sant Pau (IIB SANT PAU), 08041 Barcelona, Spain; crodriguezs@santpau.cat; 4CIBER de Enfermedades Cardiovasculares (CIBERCV), Instituto de Salud Carlos III, 28029 Madrid, Spain; 5Laboratorios Ordesa S.L., Scientific Communication Department, 08830 Sant Boi del Llobregat, Spain; 6Laboratorios Ordesa S.L., Medical Department, 08830 Sant Boi del Llobregat, Spain; 7Gastroenterology and Nutrition Service, Hospital Infantil Universitario Niño Jesús, 28009 Madrid, Spain

**Keywords:** toddlers, Mediterranean diet, growing-up milk, balanced diet

## Abstract

During the last decade, feeding patterns, more specifically those of children, have worsened—affecting dietary habits and Mediterranean diet adherence. Here, we examine the post-pandemic feeding habits of Spanish toddlers. A total of 2465 parents of children aged between 12 and 36 months completed an online 25-item multiple-choice survey asking about dietary habits and Mediterranean diet adherence. Only 34 children (1.38%) had an adequate intake of all of the food groups included in the questionnaire. Adherence worsened as toddlers grew (*p* < 0.0001). Further, lower compliance was found in children with a higher intake of fast food (*p* < 0.001), those with siblings (*p* = 0.0045), and children who were the second or third child (*p* = 0.0005). The food group with the most commonly reported adequate intake was fish (88% of children), followed by pulses (80%), water (79%), and meat (78%). Cow’s milk was the most commonly consumed dairy product among all age groups analyzed. Half of the children exhibited a low consumption of milk and dairy products. These results showed that a lack of adherence to a balanced diet is common among Spanish toddlers in the post-pandemic period and that greater parent education could improve the nutrition of toddlers.

## 1. Introduction

Early childhood is a period of rapid development, and with that comes a long list of nutritional requirements. The feeding of children aged 12 to 36 months (toddlers) has changed over the last 10 years as nowadays, their diets more closely resemble that of older children [1]. Toddlers can eat by themselves with a spoon from 12 months of age and with a fork from 30–36 months. They usually play while eating and clearly show their preferences and dislikes for specific foods. While it may seem that the feeding of these children is insufficient, it should be considered that their energy requirements are lower than older children, which explains their smaller total intake. Although food intake can vary depending on the day, healthy toddlers have a stable weekly caloric consumption [1]. A toddler’s diet, as is the case with the rest of the population, should be balanced and inclusive of the appropriate nutrients and micronutrients. The Mediterranean diet has been internationally used as the model for a healthy and balanced diet for more than 50 years [2].

The traditional Mediterranean diet is characterized by a high intake of plant foods (vegetables, fruits, pulses, bread, cereals, and nuts), the use of olive oil, seasonal and fresh products, a high intake of dairy products, and sporadic red meat consumption. However, the Mediterranean diet model varies by country; therefore, there is no unique single Mediterranean diet profile. In addition, feeding patterns in Mediterranean countries have gradually shifted overtime, now including more highly processed foods, foods with a low nutrient density and fiber content, and foods that are rich in saturated fatty acids and sugars [3]. In a systematic review and meta-analysis of 18 studies among children and teenagers based in Spain, Italy, Greece, Turkey, and Cyprus, the adherence to the Mediterranean diet varied markedly. Importantly, the results evidenced that the Mediterranean diet is becoming increasingly abandoned [4]. Indeed, in a recent study of European adults, adherence to the Mediterranean diet ranged from moderate to weak [5]. Despite the importance of a balanced diet in toddlers, very few studies have analyzed feeding patterns in European children aged between 12 and 36 months. 

According to a 2017 study by the Global Burden of Disease, an unbalanced diet has a considerable impact on mortality and morbidity [6]. To avoid long-term health complications, healthy feeding habits should be initiated early in life, within the family. A healthy diet during childhood is of paramount importance since it protects from micronutrient deficiency, obesity, and other malnutrition disorders, as well as from non-infectious diseases [7]. Despite growing evidence of the benefits of a balanced diet, feeding habits are steadily worsening. 

Unfortunately, in 2020, the COVID-19 pandemic led to huge changes in people’s lifestyles all over the world. In Spain, to slow the spread of the pandemic, a nationwide lockdown was decreed from 13 March to 21 June 2020. With the exception of those working essential frontline jobs, during this period, people were confined to their homes, only going outside to access essential goods and services, such as food, medicine, and healthcare. Children under 14 years of age could not leave home and schools were closed [8]. The impact of the lockdown on dietary habits has been studied in adults [9], children, and teenagers with eating disorders [10]; however, this issue has been ignored amongst toddlers, at least in Mediterranean countries. 

The present study was performed just over a month after the lockdown ended and coincided with the school holidays, a period during which children were still at home and/or with their parents and caregivers. We investigated the feeding habits of Spanish toddlers just after the pandemic, specifically their adherence to a Mediterranean diet, and aimed to identify potential nutritional deficiencies and establish whether the age of toddlers had any influence over the results. 

## 2. Materials and Methods

The study consisted of a web-based questionnaire regarding the adequacy of feeding patterns and Mediterranean diet adherence, as well as some potentially related factors.

### 2.1. Participants

Participants were parents or caregivers of family units who were registered members of the Spanish Club of Families (Club Familias; www.clubfamilias.com, accessed on 20 July 2020), a web platform developed by a private company aimed at promoting healthy habits in different age groups of the population. The process of collecting data from users of the web platform who had agreed to participate in the present study legally complied with Spanish data protection guidelines. The club database was searched to identify families with children aged between 12 and 36 months. 

### 2.2. Survey Questionnaire

A 25-item multiple-choice questionnaire composed of three sections: family profile (items 1 to 5), previous and current milk type consumed by toddlers (items 6 to 8), and feeding habits (items 9 to 25) was developed and sent to more than 120,000 families. In July 2020, all selected families received an e-mail with an explanation of the survey and a link to the questionnaire. A reminder was sent after 15 days. Questionnaires had to be completed online within a month from receipt of the e-mail and answered questionnaires were included in a database. Participation was voluntary, anonymous, and non-remunerated. 

Valid questionnaires were previously defined per protocol as those from families with children aged between 12 and 36 months. Questionnaires wherein some of the items were not completed were accepted only when the number of empty items was very low. Once the survey period ended, valid questionnaires were processed. Moreover, results were classified into three groups: children aged 12 to 18 months, children aged 19 to 24 months, and children aged 25 to 36 months. 

### 2.3. Assessment of Adherence to Adequate Intake 

Adherence to adequate feeding patterns was assessed. For this purpose, the adequate intake of each different food group was established based on nutritional recommendations for a balanced Mediterranean diet [11,12,13] (Table 1). Each food group was scored: adequate = +1 point, inadequate = 0 point.

Researchers subsequently developed a scale to score the number of registered adequate intakes from 0 (worse adherence) to 12 (best adherence) with the sum of all food groups (adequate intake score). The level of adherence was expressed as the percentage of children that had an adequate intake of each food group. Adequate intake was further analyzed by age, family profile, number of siblings, order of siblings, type of milk consumed by the toddler, duration of breastfeeding, sex, number of meals shared with adults, frequency of fast food intake, and geographical area. For study purposes, Spain was divided into 3 zones, north, center, and south. 

### 2.4. Statistical Analysis

A descriptive analysis was performed. Categorical variables were expressed with frequency and percentage, whereas continuous variables were reported using measures of central tendency. The results for the three age groups considered were compared to determine possible differences in the answers to the questionnaire. The Fisher exact test was used to analyze differences between categorical variables. Continuous variables were compared using the *t* test (two groups) and the ANOVA test (three or more groups). All statistical analyses were conducted using SAS/STAT^®^ (SAS Institute, Inc., Cary, NC, USA) version 9.4, and *p* values < 0.05 were considered statistically significant.

## 3. Results

### 3.1. Survey Results

The questionnaire was sent to 121,665 families, from whom 2465 valid questionnaires were returned. All of the results in this document are based on valid answers. By geographic areas, 797 (32.46%) questionnaires came from the north of Spain (Álava, Asturias, Barcelona, Burgos, Cantabria, La Coruña, Girona, Guipúzcoa, Huesca, La Rioja, León, Lleida, Lugo, Navarra, Orense, Palencia, Pontevedra, Soria, Tarragona, Vizcaya, Zaragoza), 941 (38.33%) questionnaires from the center (Alicante, Ávila, Baleares, Cáceres, Castellón, Ciudad Real, Cuenca, Guadalajara, Madrid, Salamanca, Segovia, Teruel, Toledo, Valencia, Valladolid, Zamora), and 717 (29.21%) from the south (Albacete, Almería, Badajoz, Cádiz, Ceuta, Córdoba, Granada, Huelva, Jaén, Las Palmas, Málaga, Melilla, Murcia, Santa Cruz de Tenerife, Sevilla). 

#### 3.1.1. Family Profile 

The characteristics of family profiles are summarized in Appendix A. Almost 90% of children were from a traditional hetero-parental family model. Most were the only child, 18% had one sibling, and less than 5% had three or more siblings; the mean number of siblings was 0.61 ± 0.97. Furthermore, 79% of children were the eldest or only child. More than half of the children (53%) were males. Most children (70%) were between 12 and 24 months old, and the mean age was 21.71 ± 6.75 months. The mean length at birth was 49.96 ± 3.46 cm, and mean weight was 3.19 ± 0.54 kg.

#### 3.1.2. History of Breastfeeding and Different Milk Intake

The survey asked questions on breastfeeding duration, bottle-feeding introduction, and milk type consumption (Table 2). During the first year of life, 15% of children were not breastfed, and 27% were breastfed for less than 6 months. The mean duration of breastfeeding was 9.2 ± 8.47 months. 

Infant formula was consumed from birth in 21% of children and before 6 months in 49%. The mean age of infant formula introduction was 3.98 ± 4.52 months. Regarding milk type consumption, the most commonly consumed type was cow’s milk, followed by growing-up milk and breast milk, although these data are influenced by the age of the children, as shown in Figure 1. Cow’s milk was the most consumed dairy alternative by all age groups. However, both breast milk and growing-up milk were more frequently consumed in those younger than 18 months than in children older than 24 months (*p* = 0.0001). 

#### 3.1.3. Feeding Patterns and Frequency of Different Food Groups Intakes

Feeding patterns and food group intakes are shown in Table 3. Almost 80% of parents felt that their children eat the correct quantity of food, although this assumption was most frequently ascribed to younger children, and older ones were commonly defined as bad eaters (*p* < 0.001). More than half of the children shared meals with adults every day at lunch and/or dinner, especially older children (*p* < 0.001 vs. younger children).

The frequency of intake of the different food groups is also shown in Table 3. Three glasses of water a day was the most common water intake, although older children drank more water than younger ones (*p* < 0.001). 

Regarding the high protein food group, meat was eaten three or four times a week by 50% of children, with no statistically significant differences between age groups, while fish was eaten once or twice a week. Fish was more commonly consumed by children younger than 18 months (*p* = 0.016), and parents gave eggs to their children once or twice a week, with no statistically significant differences between age groups. Additionally, one or two glasses of milk a day was the intake for 65% of children, although younger children drank a higher quantity more often (*p* < 0.001). Dairy products (yogurt, cheese, and others) were eaten once or twice a day by 57% of children, although intake was lower in younger children (*p* < 0.001). The combination of milk and/or dairy products was consumed three times a day by 74% of children, with statistically significant differences between the low intake among children younger than 18 months and high intake among children older than 24 months (*p* = 0.0002).

In the high carbohydrate content foods group, cereals were consumed once a day by 57% of children, with the highest frequency being in children younger than 18 months (*p* = 0.0001), while pasta, white rice, or potatoes were eaten once or twice a week by 45% of children, especially by children younger than 18 months (*p* < 0.001). Salads and vegetables were consumed once a day by 58% of children. Younger children consumed these foods more frequently than older ones (*p* < 0.001). The frequency of consumption regarding fresh or canned fruits was at least once a day for 76% of children, although consumption decreased in children older than 24 months (*p* = 0.0001).

Concerning sugary drinks (soft drinks and processed fruit juices), up to 66% of children have never had one, with higher consumption rates being observed in children older than 24 months (*p* < 0.001). Although 23% of parents never gave their children cookies, processed baked goods, chocolate, or candies, 29% allowed them to eat these products once or twice a week. Almost half of the children (48%) have never eaten fast food; however, older children included cookies, processed baked goods, chocolate or candies, and fast food in their diet once or twice a week (*p* = 0.028).

### 3.2. Adherence to Correct Dietary Habits and the Mediterranean Diet

According to the adequate intake established in this survey analysis, the best adherence to correct dietary habits was for fish (88% of children), followed by pulses (80%), water (79%), meat (78%), and soft drinks and processed fruit juice (77%) (Table 4). Only 34 children (1.38%) had an adequate intake of all food categories. The mean punctuation for adherence to correct dietary habits and Mediterranean diet (adequate intake score) was 8.12 ± 1.92 (67.9%), meaning that the intake was adequate for 8 of the 12 food groups considered.

The adequate intake of different food groups was influenced by age. Statistically significant differences were detected for the adequate intake of water (*p* < 0.001), milk/dairy products (*p* < 0.001), fruits (*p* = 0.073), cereals (*p* < 0.001), pasta/white rice/potatoes (*p* < 0.001), meat (*p* = 0.335), fish (*p* = 0.023), and cookies/processed bakery products (*p* < 0.001). The average adherence to correct dietary habits was 69.63% in children aged 12–18 months, 68.01% in those aged 19–24 months, and 64.71% in children aged 25 months or older. The differences were statistically significant (*p* < 0.0001), with intake worsening as age increased.

Children whose parents had stated they were good eaters scored better. In contrast, children who were considered bad eaters had the worst score (*p* < 0.001). Moreover, adherence to intake patterns was better in children without siblings (*p* = 0.0045) and in those who were the eldest or only child (*p* = 0.0005).

Lower intake of fast food was related to better general adherence to correct habits. This was evidenced by the fact that an average of 70% of children who had an adequate intake of the different food groups were reported as never having consumed this type of food compared to 55% in children who consume it three or four times a week (*p* < 0.001). Differences were also found according to the geographical area, with the best adherence in the north of Spain (especially Cantabria) and the worst in the south (*p* = 0.0002).

No statistically significant differences were found between adherence with feeding patterns and the type of milk consumed, breastfeeding duration, child’s sex, or sharing meals with adults. However, there was a trend toward better adherence in children who had been breastfed for longer.

## 4. Discussion

Our study provides a picture of the feeding patterns and adherence to the Mediterranean diet in Spanish toddlers in the post-pandemic period. It is worth noting that less than 1.5% of the surveyed children had an adequate intake of the 12 food groups assessed. The low intakes reported for milk and dairy products and cereals is of particular importance. For both food groups, less than half of the children consumed them on an adequately frequent basis. These findings were unexpected because a previous Spanish study of 154 children ranging from 6 months to 30 months of age revealed an excessive milk daily intake in toddlers [14]. We could hypothesize that comments made via social media and blogs by people who are not nutrition professionals advising against milk consumption may underlie this behavior. In fact, according to the 2019 Report on Food Consumption in Spain, the intake of milk and dairy products was slightly inferior to that reported in 2018, and people with a low socioeconomic status had an especially shorter intake [15]. Causes of reduced milk and dairy product consumption should be thoroughly studied in order to design initiatives for reversing this trend. 

Interestingly, almost 90% of children were offered fish at an appropriate frequency, in contrast with the aforementioned Spanish study, which revealed excess consumption of meat, fish, and sugar-free food [15], and nearly 80% had an adequate intake of pulses. Despite their healthy nutritional value [16], the intake of nuts was not considered in our survey as their rate of consumption among the studied population was so low, likely due to the choking risk in children younger than three years old. 

The consumption of soft drinks and fruit juices was adequate in around 77% of the sample and higher in older children without arriving at a statistical significance (*p* = 0.067). In previous studies, the consumption of these kind of products in toddlers has been linked to the caregiver’s perception of them as high nutritional value alternatives for their children feeding [17], emphasizing the necessity for correct nutritional education.

Excessive consumption of protein, mainly milk- and meat-derived proteins, has been observed both in Spain [18,19,20] and in other countries [21,22,23]. The design differences in the questionnaires and studies precluded a rigorous comparison of our findings with the existing literature. However, children’s feeding patterns have been steadily changing for almost 20 years in Spain [3].

As expected, less frequent fast food intake was related to a better adherence to Mediterranean diet habits. In addition, we suggest that the differences found between the north and south of Spain could be attributed to different types of feeding or perhaps to different traditions or lifestyles [24,25]. According to a report on food intake in Spain, milk and dairy products were more frequently consumed in the north than in other areas of Spain. Similarly, different types of food were consumed more frequently in the north [14]. 

The fulfillment of nutritional requirements in toddlers was evaluated in the Spanish study ALSALMA in 2015, where parents or caregivers completed a diary on food intake by children younger than 36 months on four non-consecutive days. In 95.9% of children, protein consumption was more than twice the recommended daily allowance. In addition, deficiency of vitamins D and E, folic acid, calcium, and iodine were recorded (vitamin D in 81.7% of children aged 13–24 months and 92.1% in children aged 25–36 months) [18]. In this context, and because of the low consumption of foods such as milk and dairy products, fruit, and vegetables observed in our study, it could be hypothesized that some toddlers could also have an inadequate intake of some vitamins and minerals. Similar to the ALSALMA study [16], in our analysis we also found a lower follow-up of correct feeding habits in older children perhaps due to the greater autonomy of the child when choosing the food they eat or less pressure from the parents in establishing an adequate frequency in the different food group consumption.

Our results show that the frequency of consumption in the food group with high protein levels was within the recommended range, according to the limits set for the study, with there being no excess presence in the diet of meat, fish, eggs, or milk/dairy products. However, seeing as we have no information about the size of the rations in the studied population, we cannot be sure if the protein level was or was not excessive, as has been the case in previous scientific publications [18]. On the other hand, the consumption of saturated fats could be higher than is desirable because 43% of children ate cookies, processed baked goods, chocolate, and/or candies one to four times a week, and more than half of the toddlers had fast food at least once a week.

Further, it is difficult to estimate the intake of omega-3 fatty acids, because the questionnaire did not enquire about types of fish or vegetables, and many foods rich in these fatty acids are not usually included in a toddler’s diet.

No data on the health status of toddlers or the socioeconomic level of families were collected, although in a previous study of Spanish children, high educational attainment in at least one parent corresponded with food quality in terms of lipids [19]. Moreover, in more than 3300 children and adolescents (aged 8 to 16 years) included in the Physical Activity, Sedentarism, lifestyles, and Obesity in Spanish Youth (PASOS) study, low adherence to a Mediterranean diet was associated with more time spent in front of screens and the low educational level of parents [26]. However, the data obtained already highlight the need for interventions to improve the quality of feeding in Spanish children.

Establishing healthy habits from the first stages of life yields a large number of benefits. Adherence to Mediterranean diets during childhood and adolescence improves not only the child’s nutritional status but also their quality of life and general well-being [27,28]. This long-term health influence has been demonstrated in several studies, reinforcing the positive influence a balanced diet has on decreasing the risk of chronic diseases such as cardiovascular disorders, diabetes, or obesity [29,30,31,32].In addition to increasing healthy feeding habits, another possibility for improving the nutritional status of toddlers and mitigating the risk of developing potential nutritional imbalances or deficiencies could come in the form of increasing adapted milk consumption [24,33]. In our study, 22% of toddlers consumed growing-up formulas before 36 months of age, with a higher prevalence in children younger than 18 months, which can be considered an additional source of micronutrients. In a French study of toddlers, cow’s milk intake (≥250 mL/d) was more associated with a risk of α-linoleic acid, iron, and vitamin C and D deficiencies than growing-up milk consumption (≥250 mL/d) (*p* < 0.001) [25]. Similarly, in the Growing Up Milk-Lite (GUMLi) trial [34], in which children aged between 12 and 24 months were followed for a year, those who drank a growing-up formula with a reduced protein content but partially enriched oligo elements had a lower protein and vitamin B_12_ intake and a higher iron, vitamin D, vitamin C, and zinc intake. In the present study, in which very few toddlers showed an adequate intake of all the considered food groups, children are likely to have nutritional deficiencies. Although general dietary patterns are unlikely to change with an increase in the consumption of growing-up formula [33], the regular intake of foods supplemented with vitamin D and calcium [24,35] or the inclusion of whole-grain-fortified ready-to-eat cereals [36] could prevent toddlers from experiencing vital nutrient deficiencies [1]. However, caregivers and parents should bear in mind that the composition of adapted milks can vary according to commercial brand, and not all of them are enriched in the same micronutrients.

Although we did not find statistically significant differences between adherence to correct feeding habits with sharing meals with adults, other studies have established that feeding habits are affected by parental dietary habits [37], and it has been suggested that parent-dependent factors can increase fruit and vegetable intake [38] and promote healthy breakfasts and snacks [37]. Therefore, parental education could improve children’s feeding and nutrition. That is why children-oriented initiatives, such as the Spanish government’s Program of Feeding, Nutrition, and Gastronomy for Pre-School Education [39], have been put in place. Nutrition education programs are still needed and should target not only children but also parents, caregivers, pediatricians, teachers, and other professionals [40]. As described in previous studies, that nutritional understanding of children and young adults is commonly influenced by their caregiver’s knowledge about healthy feeding habits [41]. 

A potential limitation of our questionnaire was that it asked about the frequency of intake but not about the consumed amount of food. Our study is also limited by response bias [30]. Given that the feeding and nutrition of toddlers is a somewhat delicate issue and the database used for the study is specifically focused on education about healthy habits, parents may have consciously or unintentionally altered their answers [21]. Another facet of the data that would be interesting to incorporate into future studies is the potential inclusion of questions regarding the socioeconomic standing of the participating families, as that can influence the feeding habits not only of infants but also of the rest of the family. 

## 5. Conclusions

The results of the present study could provide more information for the development and implementation of initiatives seeking to improve the feeding habits of children and their attitudes towards the Mediterranean diet. The relevance of education programs is supported by the finding that adequate intakes were less frequent in older children and toddlers with siblings, perhaps due to the parents’ heightened dedication to their first and only child, especially during the first years of their life. 

When toddlers become older and gain more control over their food choices, they consolidate their future eating behavior. Anticipatory guidance by caregivers and health care professionals could prevent the adoption of an unbalanced diet among young people, protecting them from unhealthy dietary habits at later ages, such as an obesogenic diet or disordered eating behaviors that can negatively influence their health status in the short and long term. To reinforce its significance, campaigns should be implemented to remind parents and caregivers about the benefits of a balanced diet in children of all age groups.

In conclusion, this study identified a series of deficiencies in the feeding habits of toddlers and young children in Spain—mainly a low intake of milk and dairy products and a worsening of feeding quality and loss of the Mediterranean diet model as children grow up or according to the number of siblings. Therefore, efforts should be made to improve the feeding habits of toddlers and their nutritional status. Educating people involved in the process of feeding toddlers could help Spanish toddlers to achieve a balanced and complete Mediterranean diet. 

## Figures and Tables

**Figure 1 nutrients-15-02049-f001:**
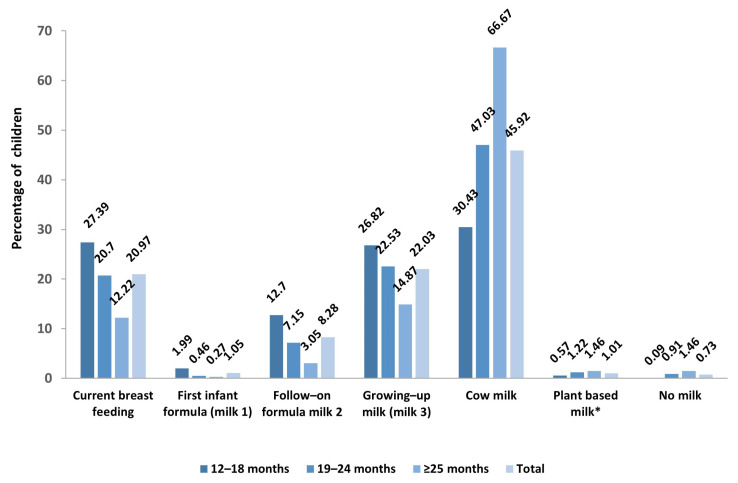
Milk type being consumed by age groups. * Oats, soy, almond, or rice.

**Table 1 nutrients-15-02049-t001:** Adequate intakes of the different food groups for a balanced Mediterranean diet according to the consensus of the study *.

Water: 3–4 times or more a day Milk + dairy products: only 3–4 times a day Vegetables/salads: once or more times a day Fruits: once or more times a day Bread/cereals: twice or three times a day Pasta, rice, potato: three or four times or more a week Pulses: once, twice, or more times a week Meat: once or twice a week and three or four times a week Fish: once, twice, or more times a week Eggs: once or twice a week Soft drinks/fruit juices: never or once or twice a month Cookies/processed baked goods: never, once or twice a month, and once or twice a week.

* Based on a UNESCO report (2010) [10], Donini et al. (2015) [11], Aranceta-Bartrona et al. (2019) [14].

**Table 2 nutrients-15-02049-t002:** Previous and current milk type consumed (n = 2465).

Item	N (%) *
Breastfeeding duration	
No breastfeeding	361 (14.92)
Less than 6 months	665 (27.49)
Between 6 and 12 months	490 (20.26)
More than 12 months	903 (37.33)
Age of infant formula introduction	
From birth	427 (20.66)
Before 6 months	1015 (49.10)
Between 6 and 12 months	413 (19.98)
Between 13 and 18 months	177 (8.56)
Between 19 and 24 months	20 (0.97)
More than 24 months	15 (0.73)
Milk type being consumed	
Breastfeeding	517 (20.97)
Infant formula (formula 1)	26 (1.05)
Follow-on formula (formula 2)	204 (8.28)
Growing-up milk (formula 3)	543 (22.03)
Cow’s milk	1132 (45.92)
Plant-based ^1^ milk	25 (1.01)
No milk	18 (0.73)

^1^ Oats, soy, almond, or rice. * Parents did not always answer all questions of the survey and therefore some items were missing.

**Table 3 nutrients-15-02049-t003:** Feeding patterns and frequency of food intake (n = 2465).

Item	N (%) *
Usually, your child eats…	
…too much	127 (5.15)
…well	1950 (79.11)
…little	307 (12.45)
…nothing or almost nothing, he/she is a bad eater	81 (3.29)
How often does your child have lunch or dinner with an adult/adults?	
Every day	1316 (53.39)
A few days a week	835 (33.87)
Never	233 (9.45)
Only at weekends	81 (3.29)
How often does your family have fast food?	
Never	1187 (48.15)
Once or twice a week	1235 (50.10)
Three or four times a week	36 (1.46)
Five or more times a week	7 (0.28)
Water (1 glass)	
Never	18 (0.73)
Once or twice a day	501 (20.32)
Three or four times a day	986 (40.00)
Five or more times a day	960 (38.95)
Milk (1 glass)	
Never	202 (8.19)
Once or twice a day	1596 (64.75)
Three or four times a day	556 (22.56)
Five or more times a day	111 (4.50)
Dairy products (yoghourt, cheese, and other) (1 cup, 1 glass, or 2 slices)	
Never	55 (2.23)
Once or twice a week	266 (10.79)
Three or more times a week	505 (20.49)
Once or twice a day	1399 (56.75)
Three or more times a day	240 (9.74)
Milk and dairy products	
Never	13 (0.53)
Once or twice a week	33 (1.34)
Three or more times a week	53 (2.15)
Once or twice a day	544 (22.07)
Three to four times a day	1184 (48.03)
Four or more times a day	24 (0.97)
Five or more times a day	614 (24.91)
Vegetables (salads and vegetables) (1 dish)	
Never	82 (3.33)
Once or twice a week	368 (14.93)
Three or four times a week	576 (23.37)
Once a day	901 (36.55)
More than once a day	538 (21.83)
Fruit (fresh or canned)	
Never	79 (3.20)
Once or twice a week	189 (7.67)
Three or four times a week	322 (13.06)
Once a day	1021 (41.42)
More than once a day	854 (34.65)
Cereals (baby food, breakfast cereals, bread) (1 bowl or 2 slices)	
Never	157 (6.37)
Once a day	1417 (57.48)
Twice or three times a day	840 (34.08)
More than three times a day	51 (2.07)
Pasta, white rice, or potatoes (1 dish)	
Never	56 (2.27)
Once or twice a week	1103 (44.75)
Three or four times a week	953 (38.66)
Once a day	353 (14.32)
Pulses (lentils, beans, chickpeas) (1 dish)	
Never	111 (4.50)
Once or twice a month	388 (15.74)
Once or twice a week	1539 (62.43)
Three or four times a week	391 (15.86)
Once a day	36 (1.46)
Meat (minced meat, red meat, chicken, and turkey) (1 fillet)	
Never	40 (1.62)
Once or twice a month	75 (3.04)
Once or twice a week	681 (27.63)
Three or four times a week	1239 (50.26)
Once a day	430 (17.44)
Fish (fish and seafood) (1 fillet)	
Never	63 (2.56)
Once or twice a month	227 (9.21)
Once or twice a week	1255 (50.91)
Three or four times a week	801 (32.49)
Once a day	119 (4.83)
Eggs (1 egg)	
Never	128 (5.19)
Once or twice a month	283 (11.48)
Once or twice a week	1690 (68.56)
Three or four times a week	323 (13.10)
Once a day	41 (1.66)
Sugary drinks (soft drinks, processed fruit juices) (1 glass)	
Never	1625 (65.92)
Once or twice a month	279 (11.32)
Once or twice a week	295 (11.97)
Three or four times a week	126 (5.11)
Once a day	140 (5.68)
Cookies, processed baked goods, chocolate, candies	
Never	558 (22.64)
Once or twice a month	443 (17.97)
Once or twice a week	727 (29.49)
Three or four times a week	335 (13.59)
Once a day	402 (16.31)

* Parents did not always answer all questions of the survey and therefore the *n* for some items could be lower than expected.

**Table 4 nutrients-15-02049-t004:** Adequate intake of the different food groups by age.

Product Type, n (%)	Age			Total	*p* Value
	12–18 mo.	19–24 mo.	≥25 mo.		
Water	753 (71.37)	549 (83.56)	644 (85.20)	1946 (78.95)	<0.001
Milk/dairy products	479 (45.40)	335 (50.99)	370 (49.14)	1184 (48.03)	<0.001
Vegetables/salads	727 (68.91)	370 (56.32)	342 (45.42)	1439 (58.38)	0.862
Fruits	831 (78.77)	512 (77.93)	532 (70.65)	1875 (76.06)	0.010
Bread/cereals	415 (39.37)	231 (35.16)	245 (32.54)	891 (36.15)	<0.001
Pasta/rice/potatoes	527 (49.95)	371 (56.47)	408 (54.18)	1306 (52.98)	<0.001
Pulses	819 (77.63)	544 (82.80)	603 (80.08)	1966 (79.76)	0.708
Meat	823 (78.01)	507 (77.17)	590 (78.35)	1920 (77.89)	0.033
Fish	946 (89.67)	577 (87.82)	652 (86.59)	2175 (88.24)	0.023
Eggs	717 (67.96)	448 (68.19)	525 (69.72)	1690 (68.56)	0.124
Soft drinks/fruit juices	932 (88.34)	486 (73.97)	486 (65.54)	1904 (77.24)	0.067
Cookies/processed baked goods	846 (80.19)	432 (65.75)	450 (59.76)	1728 (70.10)	<0.001
Average adequate intake	8.36 ± 1.84	8.16 ± 1.98	7.76 ± 1.93	8.12 ± 1.92	<0.0001

## Data Availability

The data presented in this study are available on request from the corresponding authors.

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
