# Peer review of "Post-Pandemic Feeding Patterns and Mediterranean Diet Adherence in Spanish Toddlers"

_nutrients, 2023, doi:10.3390/nu15092049_

Round 1

Reviewer 1 Report

It is a good manuscript, interesting in the field, and I congratulate the authors for it. However, in order to be published, the manuscript especially requires some additions, as well as some clarifications. Below are my observations, point by point, and related recommendations. I also attached the PDF form, my comments in text. 

IMissing some information regarding the questionnaireA properly developed questionnaire must go through the following stages in order to constitute scientific research:

1. Clear definition of research objectives and research questions: This step involves clearly defining the research objectives and research questions, which should focus on the area of interest and ensure that the questionnaire will be relevant.

2. Elaboration of a preliminary list of questions: Based on the research objectives and the research questions, a preliminary list of questions will be drawn up, covering all relevant aspects.

3. Preliminary testing of the questions: Before finalizing the questionnaire, it is recommended to test the preliminary questions on a small group of subjects, in order to identify any problems related to the understanding or interpretation of the questions.

4. Revision of the questions: Based on the results of the preliminary test, the questions will be revised and adapted according to the feedback received.

5. Ensuring the validity and fidelity of questions: Validity and fidelity are two essential characteristics of a questionnaire. Validity refers to the extent to which the questionnaire questions accurately measure what is intended to be measured, while fidelity refers to the extent to which the questions are stable and consistent. To ensure the validity and fidelity of the questions, it is recommended to use standardized methods for developing and testing the questions.

6. Pre-testing the questionnaire: After the questionnaire has been developed, it is recommended to pre-test it on a sample of participants to identify any problems related to the understanding or interpretation of the questions.

7. Data analysis and revision of the questionnaire: After data collection, it is recommended to analyze them to identify possible problems or deficiencies of the questionnaire. Depending on these, the questionnaire can be revised and adapted.

8. Detailed reporting of the questionnaire development and validation process: To ensure the transparency and reproducibility of the research, it is important to include in the research report details about the questionnaire development and validation process.

The authors have to mention how the questionnaire was established. Here are some recognized bibliographic sources in scientific research that address the development of a questionnaire:

-       Fowler Jr, F. J. (2013). Survey research methods. Sage publications. ISBN: 978-1-4522-0365-8https://www.worldcat.org/title/survey-research-methods/oclc/918559564

-       Hair Jr, J. F., Black, W. C., Babin, B. J., & Anderson, R. E. (2014). Multivariate data analysis (7th ed.). Pearson Education Limited. ISBN: 978-1-292-02190-4

-       Krosnick, J. A. (1999). Survey research. Annual review of psychology, 50(1), 537-567. DOI: 10.1146/annurev.psych.50.1.537https://www.annualreviews.org/doi/abs/10.1146/annurev.psych.50.1.537

-       Leeuw, E. D., & Hox, J. J. (2008). Introduction to multilevel analysis. Sage publications. https://psycnet.apa.org/record/2008-02337-012

-       Oppenheim, A. N. (2000). Questionnaire design, interviewing and attitude measurement. Continuum. ISBN: 978-0-8264-4736-5 https://dimas0709.files.wordpress.com/2018/02/a-n-oppenheim-questionnaire-design-interviewing-and-attitude-measurement-1992.pdf

-       Tourangeau, R., & Yan, T. (2007). Sensitive questions in surveys. Psychological Bulletin, 133(5), 859-883. DOI: 10.1037/0033-2909.133.5.859 https://psycnet.apa.org/record/2007-12463-007

II. The article is required to present a distinct section of Conclusions. In its absence, the research has no finality. The conclusions must be so detailed as to give the research a sufficiently consistent importance to be able to validate the research itself.

III. The list of bibliographic references should be expanded to confirm the information provided by the article, in the sense that it should present more research already published in similar/related fields, even specific to other countries. Here are some suggestions that would improve the quality of the research, by presenting the information and citing it:

-       Balan, I.M.; Gherman, E.D.; Gherman, R.; Brad, I.; Pascalau, R.; Popescu, G.; Trasca, T.I. Sustainable Nutrition for Increased Food Security Related to Romanian Consumers' Behavior. Nutrients 2022, 14, 4892. https://doi.org/10.3390/nu14224892

-       Harris, J.L.; Romo-Palafox, M.J.; Gershman, H.; Kagan, I.; Duffy, V. Healthy Snacks and Drinks for Toddlers: A Qualitative Study of Caregivers' Understanding of Expert Recommendations and Perceived Barriers to Adherence. Nutrients 2023, 15, 1006. https://doi.org/10.3390/nu15041006

-       Hernández-López, I.; Ortiz-Sola, J.; Alamprese, C.; Barros, L.; Shelef, O.; Basheer, L.; Rivera, A.; Abadias, M.; Aguiló-Aguayo, I. Valorization of Local Legumes and Nuts as Key Components of the Mediterranean Diet. Foods 2022, 11, 3858. https://doi.org/10.3390/foods11233858

-       Balan, I.M.; Gherman, E.D.; Brad, I.; Gherman, R.; Horablaga, A.; Trasca, T.I. Metabolic Food Waste as Food Insecurity Factor—Causes and Preventions. Foods 2022, 11, 2179. https://doi.org/10.3390/

Author Response

We thank the reviewer for his/her favorable comments and for his/her criticisms that have certainly helped to increase the quality of the manuscript.

Please see the attachment to consult our answers to them.

Reviewer 2 Report

This nice research article examines the post-pandemic feeding habits of Spanish toddlers. 2465 parents of children aged between 12 and 36 months completed an online 25-item multiple-choice survey asking about dietary habits and Mediterranean diet adherence.

Authors concluded that low adherence to balanced dietary habits is common among Spanish toddlers in the post-pandemic period. Parent education could improve toddlers’ nutrition.

·      The major concern I have is related to population selection. The authors declare that they have submitted the questionnaire to families units registered in the Spanish Club of Families (Club Familias; www.clubfamilias.com), a web platform aimed at promoting healthy feeding.

The study was therefore conducted on families already aware of the importance of nutrition in infants, therefore a selected and well-informed population. This can be deduced, if not, please specify how the registration to the Spanish Club of Families takes place in the methods. If, on the other hand, they are well aware families, this aspect must be discussed within the limitation of the study section.

·      Since it is known from the literature that atherosclerosis already begins in the early stages of life, it would be appropriate to include a reference to this in the discussion to reinforce the usefulness of a good adherence to the Mediterranean diet (see D'Ascenzi F, et al. When should cardiovascular prevention begin? The importance of antenatal, perinatal and primordial prevention. Eur J Prev Cardiol. 2019 Dec 13:2047487319893832. doi: 10.1177/2047487319893832.

·      I agree with the authors that the lack of information on socioeconomic status is a shortcoming but this is well described in the manuscript.

It would also be important to define the geographical area of the families who responded, because Spain is very large and adherence to the Mediterranean diet can vary in different areas (by the sea, inland, mountains, etc.)

Author Response

(The authors gave the same response as above.)

Reviewer 3 Report

The study “Post-pandemic feeding patterns and Mediterranean diet in Spanish toddlers” of   Ana Isabel Reyes-Domínguez  et al  provides a picture of the feeding patterns and adherence to the Mediterranean diet in Spanish toddlers in the post-pandemic period. The Mediterranean diet was increasingly abandoned  and in particular in pandemic and post pandemic period

The results of the present study could give more information for the development and implementation of initiatives to improve the feeding habits of children and their approach to the Mediterranean diet. Relevance of education programs is supported by the finding that adequate intakes were less frequent in older children and in toddlers.

The study present a potential limitation of questionnaire that asked the frequency of  intake but not about the consumed amount of food. The study is also limited by response bias. In future studies is to include the socioeconomic information of the families that can influence the feeding habits not only of infants but also of the rest of the members. The study while not original but very interesting because  identified a series of deficiencies in the feeding habits of  toddlers and young children in Spain. Therefore, efforts should be made to improve toddlers’ feeding habits and nutritional status. 

Author Response

(The authors gave the same response as above.)

Round 2

Reviewer 1 Report

The manuscript has been improved and now reaches the scientific level required by the journal for publication. However, in my opinion, the Conclusions could be expanded, because the presented research offers much more results, leading to conclusions, than those presented in the Conclusions.

1. The paragraph from Discussions "The results of the present study could give more information for the development and implementation of initiatives to improve the feeding habits of children and their approach to the Mediterranean diet. Relevance of education programs is supported by the finding that adequate intakes were less frequent in older children and toddlers with siblings perhaps due to the higher parents' dedication to their first and only especially during the first years of life. Campaigns should be implemented to remind parents and caregivers of the benefits of a balanced diet in children of all age groups." I think it would be more appropriate to be presented in the Conclusions.

2. At the same time, I recommend that the authors estimate in the Conclusions what will be the implications of the deficiencies in the feeding habits of toddlers and young children in Spain, respectively how the authors consider that these deficiencies will impact in the future, in the event that measures are not adopted remedy.

Both are just suggestions, which I leave to the Editor's decision.

Author Response

We thank the reviewer for the additional minor revisions to improve our manuscript quality all of which have been implemented.

In the attached document we answer to the different comments.
